# Cardiac Development Long Non-Coding RNA (*CARDEL*) Is Activated during Human Heart Development and Contributes to Cardiac Specification and Homeostasis

**DOI:** 10.3390/cells13121050

**Published:** 2024-06-18

**Authors:** Isabela T. Pereira, Rubens Gomes-Júnior, Aruana Hansel-Frose, Rhaíza S. V. França, Man Liu, Hossam A. N. Soliman, Sunny S. K. Chan, Samuel C. Dudley, Michael Kyba, Bruno Dallagiovanna

**Affiliations:** 1Basic Stem Cell Biology Laboratory, Instituto Carlos Chagas-FIOCRUZ-PR, Curitiba 81350-010, PR, Brazil; rubens_gjr@hotmail.com (R.G.-J.); aruanahansel@gmail.com (A.H.-F.); rhaizasvf@gmail.com (R.S.V.F.); bruno.dallagiovanna@fiocruz.br (B.D.); 2Department of Medicine, Division of Cardiology, University of Minnesota, Minneapolis, MN 55455, USA; liu00290@umn.edu (M.L.); sdudley@umn.edu (S.C.D.J.); 3Lillehei Heart Institute, University of Minnesota, Minneapolis, MN 55455, USA; nagyx051@alumni.umn.edu (H.A.N.S.); sschan@umn.edu (S.S.K.C.); kyba@umn.edu (M.K.); 4Department of Pediatrics, University of Minnesota, Minneapolis, MN 55455, USA; 5Stem Cell Institute, University of Minnesota, Minneapolis, MN 55455, USA

**Keywords:** cardiomyocytes, lncRNA, gene expression, cardiac development

## Abstract

Successful heart development depends on the careful orchestration of a network of transcription factors and signaling pathways. In recent years, in vitro cardiac differentiation using human pluripotent stem cells (hPSCs) has been used to uncover the intricate gene-network regulation involved in the proper formation and function of the human heart. Here, we searched for uncharacterized cardiac-development genes by combining a temporal evaluation of human cardiac specification in vitro with an analysis of gene expression in fetal and adult heart tissue. We discovered that *CARDEL* (CARdiac DEvelopment Long non-coding RNA; LINC00890; SERTM2) expression coincides with the commitment to the cardiac lineage. *CARDEL* knockout hPSCs differentiated poorly into cardiac cells, and hPSC-derived cardiomyocytes showed faster beating rates after controlled overexpression of *CARDEL* during differentiation. Altogether, we provide physiological and molecular evidence that *CARDEL* expression contributes to sculpting the cardiac program during cell-fate commitment.

## 1. Introduction

The delivery of oxygen and nutrients to the many tissues throughout the body is possible due to the function of a complex muscular organ, the heart. The function of the heart depends on the interplay of different cell types, including cardiomyocytes, smooth muscle cells, endothelial cells and cardiac fibroblasts [1,2]. Cardiomyocytes are the fundamental work unit of the heart; they ensure contraction of the chambers and efficient blood flow. To make a specialized cardiomyocyte is a complex process. During embryonic development, the commitment to the cardiac lineage includes several steps that are regulated by a network of transcription factors and signaling pathways that control the specification and maturation of cardiac cells [1,3,4].

Cardiac tissue is mostly derived from the mesoderm layer, which is one of the three distinct germ layers derived from the embryonic inner cell mass. Lineage-tracing and molecular-profiling experiments have provided evidence that cardiovascular progenitors are specified early in development, mostly at the stage of mesoderm induction [5]. Animal models have made important contributions to our knowledge of cardiac developmental events, especially those associated with early genetic pathways [6]. However, it has become clear that there are genomic differences among species that directly impact the regulation of development. Moreover, species differences in heart physiology, such as heart rate, calcium and potassium currents and myosin composition, make modeling human diseases in mice challenging [7,8,9]. The limited suitability of animal models to address human-specific aspects of development, disease and therapy emphasizes the importance of human-specific studies. In addition to the problem of restricted access to human fetal-derived biological material, the advantages of using human cardiomyocyte-based platforms for drug discovery, predictive toxicology and cell therapy have promoted the rapid implementation of differentiation protocols based on human pluripotent stem cells (hPSCs) [10,11,12].

The careful orchestration of several signaling pathways and regulatory genes is crucial for successful heart development and therefore deserves special attention, as defects in these early cell-fate decisions might contribute to congenital heart disease or, in severe cases, stillbirth [4,13]. One key advantage of in vitro cardiac-differentiation experiments is the ability to recapitulate key developmental events, making them readily accessible for study. In association with a variety of “omics” assays, it represents a powerful approach by which to uncover new players in cardiac specification. Here, we aimed to develop a strategy to find and test a novel cardiac developmental gene. We combined a temporal evaluation of human cardiac specification in vitro with an analysis of gene expression in fetal and adult heart tissue to search for uncharacterized genes. This strategy led us to find *CARDEL* (CARdiac DEvelopment Long non-coding RNA), which we functionally characterized using genetic approaches. *CARDEL* knockout and controlled overexpression in hPSCs were accomplished using two distinct and independent cell lines. These complimentary approaches showed that *CARDEL* is required for successful cardiac specification as well as for final cardiomyocyte physiological function, establishing *CARDEL* as a critical cardiac gene.

## 2. Materials and Methods

### 2.1. Cell Culture and Cardiomyocyte Differentiation

Human embryonic stem cells, H1 line, were obtained from WiCell Research Institute (Madison, WI, USA) under a Materials Transfer Agreement (No. 18-W0416) with Carlos Chagas Institute. A human induced pluripotent cell line, IPRN13.13, was generated as described by Darabi et al., 2012 [14]. The H1 hESC line and IPRN13.13 iPSC line are male and present normal karyotypes. Cells were cultured on Matrigel-coated dishes using either mTeSR-1 or TeSR-E8 medium (Stem Cell Technologies, Vancouver, BC, Canada). The medium was replaced daily until the cells reached 70–80% confluence. Cells were maintained in a 37 °C, 5% CO_2_ incubator. 

Both cell lines were subjected to a protocol based on that by Lian et al. [15], with a few modifications. Briefly, 5 × 10^5^ cells/well were seeded in a Matrigel-coated 12-well plate in mTeSR-1 + 10 µM Y27632; this time point corresponds to day 3 of differentiation. The medium was replaced with 2 mL of mTeSR-1 per well on days 2 and 1. On day 0, 2 mL of RPMI/B-27 minus insulin medium supplemented with 12 µM CHIR99021 were added to each well. After 24 h (day 1; mesoderm stage), the medium was replaced with 2 mL of RPMI/B-27 minus insulin. On day 3 of differentiation, 1 mL of conditioned medium from the well was mixed with 1 mL of fresh RPMI/B-27 minus insulin medium and supplemented with 10 µM XAV939 to be replaced in the well. On day 5, the medium was replaced with 2 mL of fresh RPMI/B-27 minus insulin; and on day 7, day 10 and day 13, the medium was replaced with 2 mL of RPMI/B-27. When the cells were maintained until day 30 of differentiation, the medium was replaced with RPMI/B-27 every 3–4 days. Beating areas were expected around day 10. In the doxycycline-inducible lines, 1 µg/mL doxycycline was added on day 5 and day 7 to complete a day 5-to-10 induction.

Beating areas were recorded on day 15 (D15) using AxioVs40 V4.8.2.0 on an inverted microscope (Carl Zeiss MicroImaging, Jena, Germany) from two distinct areas within the same well with three wells per differentiation replicate, for three (hiPSC) or two (hESC) differentiation replicates in total. Beating area and rate were quantified using motion-tracking software [16].

### 2.2. Lines with Doxycycline-Inducible Expression and Knock-Out Lines

For the lines with doxycycline-inducible expression, the *CARDEL* sequence (4.5 kb) was amplified from the DKFZp686D0853 clone (gift from Dr. S. Weimann, German Cancer Research Center, Heidelberg, Germany) and cloned first into the pDONOR221 vector, then into the AAVS1-TRE-GW-rtTA vector through the Gateway methodology. hPSCs were transfected using 600 ng of AAVS1-TRE-GW-*CARDEL*-rtTA construct, 200 ng of ZFN(R) and ZFN(L) for the insertion of the doxycycline-inducible system into the AAVS1 locus. Positive transfected cells were selected with 0.2 µg/mL puromycin for 7 days and single-colony cloned. Doxycycline-inducible expression was confirmed by RT-qPCR.

For the knock-out lines, two guide RNAs were designed neighboring the *CARDEL* locus and were cloned into the pSpCas9(BB)-2A-RFP vector. hPSCs were transfected using 500 ng of each construct. Mutated colonies were screened by PCR after single-colony cloning. Genotyping was performed using genomic DNA and primers for the *CARDEL* internal sequence (WT-F: GCTGGGCAGGAACCTTACAA, WT-R: TGTTTGAGCCGAAGGAGCAT; detection of a 780 bp wild-type allele) and for the *CARDEL* neighboring sequences (KO-F: GGGCCTTTGACTACAAATGGAT, KO-R: GCAGAGGAATGTGGAAGGCT; detection of a 514 bp deleted allele and 10,500 bp wild-type allele).

### 2.3. Karyotype

The new derived cell lines were seeded in six-well plates and incubated with 0.32 µg/mL colchicine for 90 min when the cells reached 90% confluence. After they had been washed with 1× PBS, the cells were incubated with 1× PBS for 10 min at 37 °C and then detached using a scraper. Then, the cells were collected and centrifuged at 800× *g* for 5 min. The cell pellet was resuspended in 57 mM KCl for a 10 min incubation at 37 °C. Cells were then fixed using a solution containing 75% methanol and 25% acetic acid for 10 min at −20 °C then washed twice in a solution of 66.6% methanol and 33.3% acetic acid. Fixed cells were used to prepare slides and for karyotype analysis.

### 2.4. Flow Cytometry

hPSC-derived cardiomyocytes were dissociated using trypsin-EDTA (0.05%) and fixed using 4% paraformaldehyde and 90% methanol. Fixed cells were incubated with 0.5% PBS/BSA, 0.1% Triton-X and 1:100 primary antibodies for cTnT (cardiac isoform Ab-1 mouse, Thermo Scientific™, Waltham, MA, USA, cat. #MS-295-P0). After washing, cells were incubated with 0.5% PBS/BSA, 0.1% Triton-X and 1:1000 secondary antibodies. Analyses were carried out using a FACSCanto II flow cytometer (BD Biosciences, Franklin Lakes, NJ, USA and FlowJo software (v. 10.7).

### 2.5. RNA Isolation and RT-qPCR

The total RNA of cells at distinct timepoints of differentiation was isolated using TRIreagent (Sigma-Aldrich, Burlington, MA, USA) according to the manufacturer’s instructions. cDNA synthesis was performed using ImProm-II^TM^ Reverse Transcriptional System (A3800—Promega, Madison, WI, USA) with 1 µg of RNA, and RT-qPCR reactions were performed using GoTaq^®^ qPCR Master Mix (A6002—Promega, Madison, WI, USA). Analyses were carried out in the QuantStudio^TM^ (Carlsbad, CA, USA) 5 Real-Time PCR System. GAPDH or POLR2A expression was used as an endogenous control.

### 2.6. Whole Cell Patch Clamp

Action potentials from day-30 cardiomyocytes were measured using the whole-cell patch-clamp technique in current clamp mode at room temperature. Pipettes (2–3 MΩ) were filled with a pipette solution containing (in mM): NaCl 5, KCl 150, CaCl2 2, HEPES 10, EGTA 5 and MgATP 5 (adjusted to pH 7.2 with KOH). The bath solution consisted of (in mM) NaCl 150, KCl 5.4, CaCl2 1.8, MgCl2 1, HEPES 15 and glucose 15 (adjusted to pH 7.4 with NaOH). All electrophysiological measurements were carried out with an Axopatch 200B amplifier and Axon Digitata 1320A A/D converter driven by a pCLAMP system (Digidata A/D and D/A boards and pCLAMP 9.2, Molecular Devices, Sunnyvale, CA, USA). Data were analyzed with Clampfit. All records were filtered with cutoff frequencies designed to avoid aliasing and were digitized at speeds at least five times the filter cutoff frequency, generally 2 kHz and 10 kHz, respectively. The APs were recorded under pulses of 3 ms in duration, 1.2-fold the threshold intensity at a stimulation rate of 0.5 Hz. Analog and P/4 methods were used for leak and capacity transient cancellation. Series resistance was partially compensated by feedback circuitry.

### 2.7. RNA-seq and Data Analysis

RNA-seq data for in vitro cardiac differentiation were previously generated by our group [17] and are available at NCBI Sequence Read Archive, accession number SRP150416. RNA-seq data heart tissues were acquired from European Nucleotide Archive (ENA) (http://www.ebi.ac.uk/ena, accessed on 25 May 2022). The accession numbers for the adult-heart-tissue data are ERR315356, ERR315430, ERR315367 and ERR315331. The project number for the fetal-heart-tissue data os PRJEB27811 [18]. Quality control, read mapping and quantification was performed using the nf-core/rnaseq (v.3.8.1) pipeline. Reads were mapped to hg38/GRCh38 and quantified with Salmon using the Gencode v38 annotation. Significantly differentially expressed genes were identified using DESeq2 (v.1.24.0) [19] with an adjusted *p*-value cutoff of 0.05 and a log2FoldChange cutoff of 2.

Polysome-bound RNA-seq data for *CARDEL* overexpression were generated using the i*CARDEL*^OE^ hiPSC cell line under control (−DOX) and induced (+DOX, 1 µg/mL doxycycline added on day 5 and day 7 to complete a day-5-to-10 induction) conditions on day 10 and day 15 of cardiac differentiation, according to a previously established protocol [17]. Quality assessment and trimming of reads were done with FastQC and Trim Galore (v.0.4.0) [20]. Reads were mapped to hg38/GRCh38 with HISAT2 (v.2.1.0) [21] and counted with HTSeq (v.0.11.1) [22]. Significantly differentially expressed genes were identified with DESeq2 (v.1.24.0) [19] with an adjusted *p*-value cutoff of 0.05 and a log2FoldChange cutoff of 1.5. Normalization of reads by RPKM was performed with edgeR (v.3.40.0) [23].

ChIP-seq data is described in [24] and is available at Gene Expression Omnibus under accession number GSE35583. BigWig files were retrieved from ENCODE (ENCODE4 v1.5.1 GRCh38 processed data).

Gene Ontology (GO) analysis was performed using gProfiler [25].

### 2.8. Data and Statistical Analysis

Graphed data are expressed as mean ± SD. Statistical analysis was performed with Prism 8.0 software (GraphPad Software, La Jolla, CA, USA). For comparison among multiple groups, a one-way ANOVA followed by Dunnett’s multiple comparison test was performed as appropriate. Comparisons between two mean values were performed with a Student’s unpaired *t*-test. A *p*-value lower than 0.05 was considered statistically significant. For RNA-seq data, DESeq2 (v.1.24.0) [19] was used for statistical analysis with an adjusted *p*-value less than 0.05 considered significant. Recording of patch-clamp data was performed by a collaborator who was blinded to the experimental conditions.

## 3. Results

### 3.1. CARDEL Is Expressed during the Human Cardiac-Lineage Specification

Taking advantage of our previously published RNA-seq dataset [17], we narrowed our search for cardiac developmental genes, looking at the genes that were differentially expressed between the mesoderm and cardiac progenitor (CP) stages of the in vitro differentiation (IVD) of hPSC. Analysis of publicly available RNA-seq data from human fetal and normal adult heart tissues was also performed as a second step to limit the genes to those mainly expressed during heart development (Figure 1a). Genes that showed upregulation in CP cells (1354 genes, Appendix A) and were found to be enriched in fetal heart tissues (3503 genes, Appendix A) were considered potential candidates (intersection = 151 genes, Appendix A). Among those genes are well-known fetal-enriched genes such as TNNI1, which is the fetal isoform expressed in cardiac and skeletal muscle during early development [26]. As our primary goal was to find novel and uncharacterized genes, Gene Ontology analysis was used to filter out genes assigned to known biological processes (Appendix A). Following these criteria (Figure 1b and Appendix A), we chose *CARDEL*, which had no prior cardiac link, for investigation of its possible functional relevance to cardiac commitment (Figure 1b).

*CARDEL* (LINC00890; SERTM2) was named for (CAR)diac (DE)velopment (L)ong non-coding RNA. It is a transcript 4612 nucleotides long that contains three exons and a polyadenylation sequence (Figure 1d). *CARDEL* is annotated in the latest Gencode version (GRCh38.p13) as a protein-coding gene; however, the coding potential of its short ORF still needs to be proven. Coding-potential-prediction algorithms, such as PhyloCSF and CPAT, classified *CARDEL*’s transcript as non-coding [27]. On the other hand, the high conservation within its predicted coding sequence (CDS) is evidence in favor of its annotation as protein-coding.

To further validate our observations with the RNA-seq data, we tested additional adult heart-tissue samples for *CARDEL* expression by RT-qPCR and found undetectable expression in those samples (Appendix A). Corroborating these results, we also interrogated a recently published database of over 900 RNA-seq samples [28], which showed that *CARDEL* expression was expressed in fetal-like cells compared to cells of adult heart tissue.

We therefore validated *CARDEL* expression in hPSC-derived cardiac-lineage cells. We examined *CARDEL* temporal expression during cardiogenesis of both hESCs and hiPSCs, using one line of each (Figure 1e), and found that *CARDEL* expression coincides with commitment to the cardiac lineage, increasing from the cardiac mesoderm (D5) to cardiomyocytes (D15) (Figure 1f). Additional analysis of public ChIP-seq datasets [24] revealed an increase in the active chromatin marker H3K4me3 on the *CARDEL* promoter during the IVD of hESC (Figure 1g), in agreement with the observed increase in expression in our data. *CARDEL* transcripts are not nuclear-enriched, as suggested by cellular fractionation and the nuclear-to-cytoplasmic ratio (N/C) of RNA abundance (Appendix A). *CARDEL* showed a similar N/C to *NORAD*, a cytoplasmic lncRNA, and a lower N/C than *NEAT1*, a nuclear lncRNA.

Together, these results suggest that *CARDEL* is specifically expressed in early cardiac-lineage cells. We therefore investigated the functional role of *CARDEL* during human cardiac specification.

### 3.2. Cardiomyocyte Differentiation and Homeostasis Were Impaired in CARDEL^KO^ Cells

The functional activity of a genomic locus can be investigated at several levels: (1) DNA, through, for example, examination of regulatory sequences (e.g., promoters and enhancers); (2) RNA, through, for example, examination of non-coding roles either in cis or in trans; and (3) protein. Considering that the *CARDEL* locus has been uncharacterized to date, we first aimed to assess whether the transcribed *CARDEL* sequence was necessary for cardiac specification. To this end, we established two *CARDEL* knock-out (*CARDEL*^KO^) lines (H1 hESC and IPRN13.13 hiPSC) using CRISPR/Cas9 technology (Figure 2a). Cleavage in both gRNA target sequences could be detected by PCR using primers up and downstream of those sequences (KO-F/R), as well as internal primers (WT-F/R) (Figure 2a,b). *CARDEL*^KO^ lines showed normal karyotypes, typical pluripotent colonies and expression of pluripotent genes, indicating that the lack of *CARDEL* expression does not affect pluripotency or genome stability (Appendix A). In addition, depletion of *CARDEL* expression was confirmed by RT-qPCR when *CARDEL*^KO^ cells were subjected to the cardiac IVD assay (Appendix A).

The potential of *CARDEL*^KO^ cells to undergo differentiation to the mesoderm lineage, the first step of cardiac differentiation, was not compromised, as those cells showed similar levels of mesoderm induction on D1 of IVD to wild-type (WT) cells (Figure 2c,d). On the other hand, *CARDEL*^KO^ hiPSC-derived cells showed impairment of cardiac differentiation, as suggested by lower numbers of cTnT-positive cells compared to WT hiPSC-derived cells. Differentiated beating areas also did not beat as expected, showing decreased beating rates on D15 (Figure 2e). Additionally, *CARDEL*^KO^ hESC-derived cells showed a decreased beating rate on D15, and very similar number of cTnT-positive cells when compared to WT cells (Figure 2f). It is worth noting that the baseline beating rate and cTnT-positive yield in the hiPSC and hESC WT lines are different (Figure 2e,f), and the lower capacity to derive cardiac cells from our parental hiPSC WT cells could explain the more striking deleterious phenotypes in these engineered cells. 

To investigate whether *CARDEL* KO would interfere with the differentiation into other lineages, we performed endoderm and ectoderm induction. The expression levels of gene markers was not distinct between WT and *CARDEL*^KO^ cells in either cell line tested (Appendix A), suggesting that the trilineage potential (mesoderm, endoderm, ectoderm) of *CARDEL*^KO^ cells is maintained and that *CARDEL* is required for later and more specific stages of cardiomyocyte differentiation.

The lack of or decrease in beating areas in the KO differentiating cells indicates that the absence of *CARDEL* expression impaired the cardiomyogenic differentiation of hPSCs and suggests that *CARDEL* is essential for the success of cardiogenic commitment.

### 3.3. Overexpression of CARDEL Improves hPSC Cardiomyogenic Differentiation and Alters Cardiomyocyte Functional Properties

The hypothesis that *CARDEL* plays an essential role in cardiogenic specification led us to evaluate whether its controlled overexpression (OE) during differentiation would affect final cardiac-lineage yield, especially in our parental hiPSC line, which usually shows a poor cardiac yield. Thus, doxycycline-inducible pluripotent cell lines were established (i*CARDEL*^OE^ hESC and hiPSC) to express a *CARDEL* transcript containing its three exons (Figure 3a). The engineered cells showed normal karyotypes, and *CARDEL* expression was confirmed by RT-qPCR after doxycycline induction in undifferentiated and cardiac cells (Appendix A). We tested different windows of induction and counted cTnT-positive cells as an output quantification. Interestingly, final cardiomyocyte yield was negatively affected in all but one of the tested conditions (Appendix A). One possible explanation could be that *CARDEL* expression has to be carefully controlled during cellular differentiation and that addition of exogenous *CARDEL* at the wrong time negatively affects the cardiogenic commitment. i*CARDEL*^OE^ hiPSCs subjected to the cardiomyogenic IVD and induced expression from day 5 to day 10 (D5-10) showed increased numbers of cTnT-positive cells and larger fractions of beating area compared to non-induced cells (Figure 3c). Curiously, endogenous *CARDEL* expression starts increasing at the cardiac mesoderm stage at day 5 (Figure 1f), and the induction from D5-10 mimics its physiological timeline (Figure 3b).

These results clearly indicate an improvement in cardiac-differentiation efficiency as an effect of *CARDEL* overexpression. On the other hand, i*CARDEL*^OE^ hESC-derived cardiomyocytes did not show significant differences in beating area or the number of cTnT-positive cells when they were induced with doxycycline. This might have happened because is more difficult to see improvement over the high cardiomyocyte yield (up to 80%) that non-induced hESCs usually have (Figure 3d). Moreover, both i*CARDEL*^OE^ hiPSC- and hESC-derived cardiomyocytes showed faster beating rates compared to the non-induced control (Figure 3e,f, Appendix A). 

Together, these results strongly suggest that overexpression of *CARDEL* during cardiomyocyte specification can improve the final differentiation output and alter the functional properties of those cells.

### 3.4. CARDEL Expression Contributes to Physiological and Molecular Changes during Cardiomyocyte Differentiation 

Since we observed an impressive increase in the beating rate of i*CARDEL*^OE^ cardiomyocytes, we further explored their functional properties. A whole-cell patch-clamp assay was performed on i*CARDEL*^OE^ hPSC-derived cardiomyocytes and allowed the identification of cardiomyocyte subtypes by their electrophysiological properties. After induction with doxycycline from D5-10, both i*CARDEL*^OE^ hiPSC- and hESC-derived cardiomyocytes showed altered action-potential (AP) profiles and overall longer action-potential durations (APDs) in induced cells. The APDs at 90% of repolarization (APD_90_) of the ventricular type of cells were 476.6 ± 22.8 ms in −DOX and 1169.0 ± 129.4 ms in +DOX for i*CARDEL*^OE^ hiPSC and 571.0 ± 21.5 ms in −DOX and 775.3 ± 36.9 ms in +DOX for i*CARDEL*^OE^ hESC (Figure 3g,h). Additionally, i*CARDEL*^OE^ hiPSC-derived atrial and nodal cells also showed increased APD_90_ values, and ventricular and nodal cells showed decreased maximum upstroke velocity (dV/dt_max_), suggesting altered contractility properties (Appendix A). i*CARDEL*^OE^ hESC-derived ventricular and atrial cells showed alterations in the AP amplitude. 

To better understand the phenotypic alterations observed after induction of *CARDEL* during differentiation, we performed RNA-seq of polysome-bound RNAs from i*CARDEL*^OE^ cardiac progenitors at day 10 and cardiomyocytes at day 15 under control and induced conditions. Compared to non-induced control cells, 195 genes were upregulated and 516 were downregulated on day 10. On day 15, 393 genes were upregulated and 522 were downregulated (Figure 4a and Appendix A). Altogether, the results show that a greater number of differentially expressed genes were downregulated in the induced (+DOX) conditions. However, genome ontology (GO) analysis of these sets of genes showed more general changes related to morphogenesis and development. Interestingly, the genes that were upregulated on either day 10 or day 15 were related to many cardiac-specific GO terms, such as “regulation of heart contraction”, “cardiac conduction”, “cardiac muscle tissue development” and “heart development” (Figure 4b). Some genes assigned to these terms are depicted in heatmaps for comparison of expression levels among the four conditions analyzed (Figure 4c,d). Cluster analysis shows that the expression levels of many genes that were upregulated in day 15 +DOX are more similar to those in day 10 +DOX cells than to those in non-induced cells on day 15 (Figure 4c). This suggests that *CARDEL* overexpression modulates the cardiac molecular profile strongly enough to distinguish both early (day 10) and late (day 15) cardiomyocytes from non-induced cells.

Among the upregulated genes, transcription factors critical to cardiac development, such as *NXK2-5* and *IRX3* can be found. Also, an extensive list of myosins and sarcomere structural proteins were upregulated in +DOX cardiomyocytes, including *MYH6*, *MYH7*, *MYL3*, *TNNT2* and *TNNC1*. Genes related to cardiac signaling and homeostasis were also upregulated in these cells; examples include the hormone atrial natriuretic peptide (*NPPA*) and the protease corin (*CORIN*), which converts the pro-atrial natriuretic peptide to its biologically active form (Figure 4d). 

These results provide robust evidence that *CARDEL* overexpression during cardiac-lineage commitment alters cardiomyogenic differentiation at both the physiological and the molecular levels, contributing to the distinct phenotype observed and suggesting a role for this gene in heart-contraction homeostasis during development.

## 4. Discussion

In this report, we showed that *CARDEL* expression is tightly controlled in differentiating cells and plays a crucial role in the physiology of derived cardiomyocytes.

In vitro cardiac differentiation of stem cells is an important tool for the study of the developmental mechanisms of the heart [29]. Studies using hPSC have led the way to understanding how gene expression is controlled during human development, as access to fetal-derived biological material is very limited. Analysis of mammalian stem-cell transcriptomes during in vitro cardiomyocyte differentiation revealed thousands of differentially expressed genes that are tightly and precisely controlled in terms of the timing of their expression [30,31,32,33,34,35,36]. However, while a very large set of biological data has been generated by high-throughput approaches, the lack of functional characterization hampers reliable identification of genes correlated with specific biological processes. Considering that functional annotation is critical to integrating and analyzing complex biological data [37], the characterization of critical developmental players contributes to countless fields, such as the elucidation of congenital heart diseases and the application of cellular and genetic therapies in regenerative approaches. 

Here, our main goal was to find and functionally characterize a cardiac developmental gene. Combining a temporal evaluation of human cardiac specification and tissue expression, we developed a strategy to search for genes that are exclusively expressed during differentiation. Our stringent approach identified *CARDEL* as one very interesting candidate based on its robust expression in independent and distinct in vitro differentiation protocols and cell lines (Figure 1). *CARDEL* has been reported to be expressed in other tissues, such as the endometrium and prostate, according to RNA-seq from normal and cancer tissue [38,39], but it remains mostly uncharacterized. For this reason, we chose a wider genetic approach to begin its functional characterization in the cardiac tissue. The deletion of its entire locus, including all exons and introns but excluding its promoter region, showed that *CARDEL* is required for good output of cardiac cells after differentiation (Figure 2). The importance of noncoding DNA elements (e.g., enhancers and insulators) in a specific locus is not clearly discriminated from the importance of the RNA transcript itself or from the act of transcription in a deletion-based knockout strategy [40,41]. For this reason, we performed an unbiased and independent analysis of the biological consequences of *CARDEL* transcript expression. Using a drug-inducible system to temporally control *CARDEL* expression, we showed that the *CARDEL* transcript comprising its three exons, when expressed from a completely independent locus, was able to promote cardiomyocyte differentiation and the cardiac-cell phenotype (Figure 3 and Figure 4).

*CARDEL* annotation shows high interspecies conservation in a predicted coding sequence (CDS), which classifies it as a protein-coding gene. However, its coding potential still needs to be proved. The stringent and arbitrary criteria traditionally used for prediction of protein-coding open reading frames (ORFs), e.g., 300 nucleotides or 100 amino acids as a size cutoff and the presence of an AUG start codon, have led to misannotation of many RNAs that contain small ORFs and potentially encode micropeptides [42,43]. Also, limitations of detection sensitivity and the size of the database for spectra searches in mass-spectrometry approaches have always represented challenges for the identification of small peptides. Recent and emerging advancements in bioinformatics, proteomics and transcriptomics have made the identification of new potential small ORFs possible, including in the heart [44,45,46,47,48]. To date, no peptide derived from *CARDEL* locus has been identified and biologically characterized. Additionally, the presence of an ORF does not necessarily mean that the transcript is being translated. For instance, the LINC00261 was found to be dynamically regulated during pancreatic differentiation of stem cells, being predominantly cytosolic and associated with ribosomes based on ribosome-profiling data [49]. However, disruption of each of LINC00261′s seven identified ORFs excluded the possibility that the derived peptides were required and suggested that it is the RNA that is involved in differentiation. Myosin heavy chain 7b (MYH7b) is also an interesting example of a locus encoding several potential active molecules: the protein myosin, a miRNA and a lncRNA. However, in the heart, the locus does not produce a protein due to an exon-skipping mechanism. It was recently shown that it is the lncRNA that has a regulatory role in affecting the transcriptional landscape of human cardiomyocytes, independent of the miRNA [50].

Many studies have found lncRNAs to be critical players during cardiac development [32,51,52,53,54]. Braveheart, Fendrr and Carmn are examples of lncRNAs that have already been functionally described, and loss of expression of these lncRNAs was associated with significant impairment of heart development [55,56,57]. Therefore, *CARDEL*’s RNA could be acting as part of a non-coding mechanism. Nevertheless, *CARDEL* gene product is shown here to be developmentally expressed and required for cardiac specification, in which process it likely acts in trans. *CARDEL* overexpression in hiPSCs showed a remarkable improvement in the final cardiac output (Figure 3). Additionally, the most consistently observed phenotype was the increase in the beating rate, which occurred in both cell lines studied. *CARDEL*-overexpressing cells also showed increased APD90 values and upregulation of many heart-contraction genes, such as cardiac ion channels and sarcomere structural genes. The development of mature sarcomeres and sarcoplasmic reticulum allow faster and efficient contraction, leading to better electric coupling of cardiomyocytes [13]. Therefore, these results suggest that, beyond improving differentiation, *CARDEL* also contributes to cardiomyocyte physiological and functional properties.

Cardiac hypertrophy is a pathological process that is accompanied by molecular changes that resemble those observed during cardiac development [58]. In addition, arrhythmia is usually associated with cardiac hypertrophy and other cardiac conditions. It is important to note that *CARDEL* was found to be enriched in fetal tissue and cardiac developmental staged cells (Figure 1) and that its overexpression caused an increase in the cardiomyocyte beating rates (Figure 3). A systematic study using RNA-seq data from 28 hypertrophic cardiomyopathy patients and 9 healthy donors identified *CARDEL* as upregulated in diseased tissues [59]. On the other hand, other studies comparing the left ventricles of dilated or ischemic human cardiomyopathy patients did not find *CARDEL* among the differentially expressed genes. So, it is possible that *CARDEL* expression is related to certain cardiac diseases, but this association remains inconclusive.

## 5. Conclusions

Combining a temporal evaluation of human cardiac differentiation in vitro with an analysis of gene expression in fetal and adult heart tissue, we found *CARDEL*, a poorly characterized gene. *CARDEL* is expressed in early cardiac lineage cells and is required for the success of cardiogenic commitment. Physiological and molecular differences were observed after overexpression of *CARDEL* during cardiac differentiation, and our results suggest a role of this gene in cardiac specification and homeostasis. While further research is still needed to clarify the exact molecular mechanisms underlying our observations, this study adds a new piece to the intricate gene network regulated during cardiac commitment. Ultimately, this effort enhances our knowledge about human development and helps to overcome the remaining challenges to understanding and treating heart diseases. 

## Figures and Tables

**Figure 1 cells-13-01050-f001:**
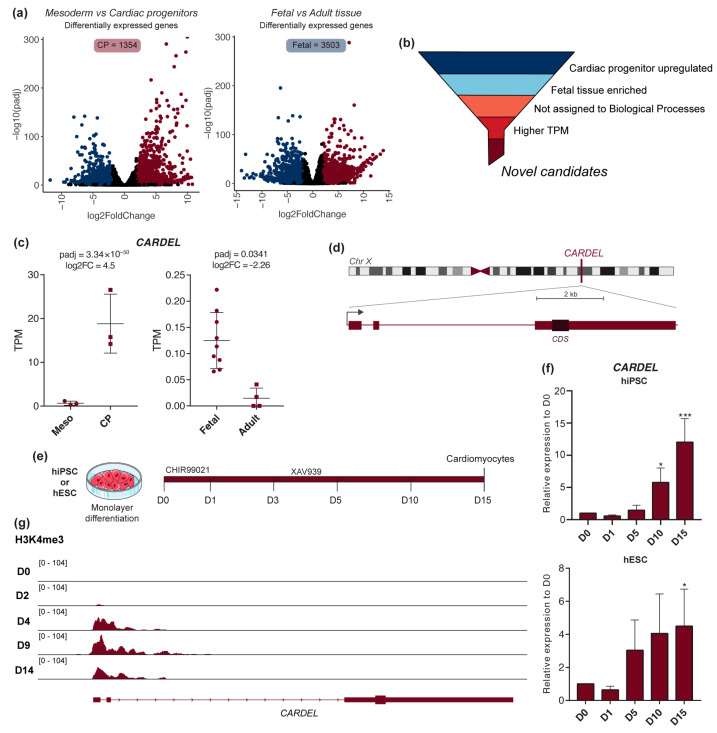
*CARDEL* is specifically expressed during the human cardiac specification. (**a**) RNA-seq analysis of mesoderm versus cardiac progenitor (CP) cells during cardiac IVD and fetal versus adult heart tissues. Volcano plots show differentially expressed genes for each comparison: downregulated genes in blue, upregulated genes in red (padj < 0.05, −2 ≥ log2FC ≥ 2). (**b**) Schematic representation of the criteria used to find candidate cardiac developmental genes. (**c**) *CARDEL* expression in mesoderm and cardiac progenitor cells during cardiac IVD (**left**) and in fetal and adult heart tissues (**right**). TPM: transcripts per million. (**d**) Schematic representation of *CARDEL*’s gene locus at chromosome X showing its 3 exons and predicted coding sequence (CDS) position. Scale bar 2 kb. (**e**) Cardiac IVD protocol used to differentiate cardiomyocytes from hiPSC or hESC. The time of collection of differentiating cells for analysis is indicated in days. (**f**) *CARDEL* expression measured by RT-qPCR in differentiating cells from hiPSC or hESC. Mean with SD; one-way ANOVA followed by Dunnett’s multiple comparisons test. Each column compared with D0, * *p* < 0.05, *** *p* < 0.001. (**g**) ChIP-seq analysis of the active chromatin marker H3K4me3 [24] on the *CARDEL* gene locus during cardiac IVD.

**Figure 2 cells-13-01050-f002:**
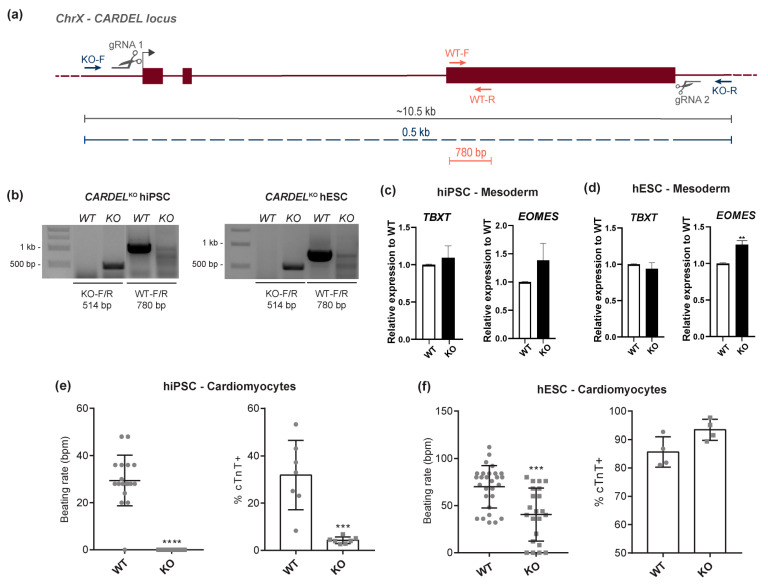
Cardiomyocyte differentiation and homeostasis were impaired in *CARDEL*^KO^ cells. (**a**) Schematic representation of the strategy used to knock out *CARDEL* from hiPSCs and hESCs using CRISPR-Cas9. Positions of the gRNAs for Cas9-directed cleavage and of the oligos for confirmation PCR are indicated, as is the expected fragment size. (**b**) PCR of genomic DNA confirming the knock-out of the *CARDEL* locus in hiPSCs and hESCs. The left gel image is the result of two cropped parts. (**c**,**d**) Expression of mesoderm markers measured by RT-qPCR on D1 (mesoderm stage) of WT or *CARDEL*^KO^ (**c**) hiPSCs and (**d**) hESCs. (**e**,**f**) *CARDEL*^KO^ (**e**) hiPSC- and (**f**) hESC-derived cardiomyocytes (D15) beating rate (**left**) and quantification of cTnT-positive cells by flow cytometry (**right**). Bpm: beats per minute. Mean with SD; Student’s unpaired *t*-test analysis: ** *p* <0.01, *** *p* < 0.001, **** *p* < 0.0001.

**Figure 3 cells-13-01050-f003:**
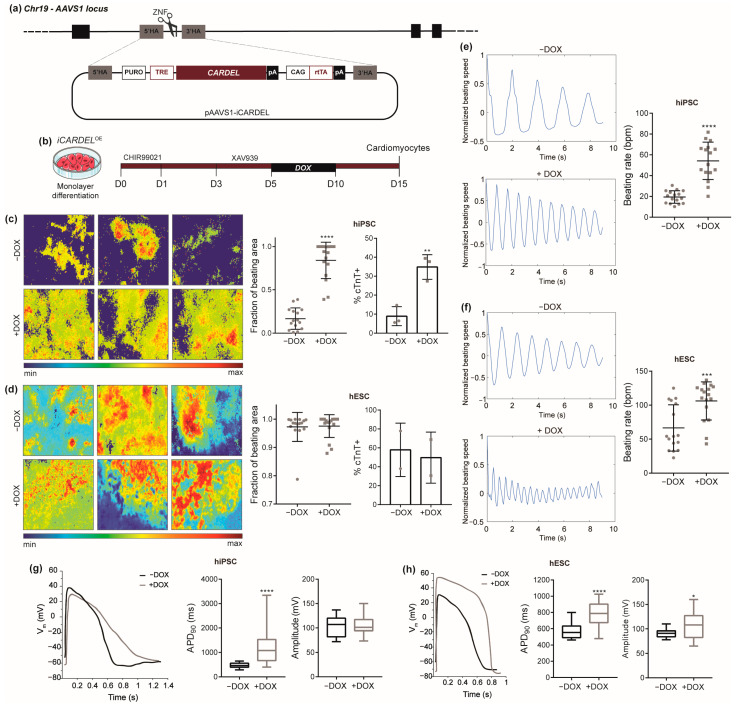
*CARDEL* overexpression improves hPSC cardiomyogenic differentiation and alters cardiomyocyte functional properties. (**a**) Schematic representation of dox-inducible system knock-in at the AAVS1 locus. (**b**) Cardiac IVD protocol used to differentiate cardiomyocytes from dox-inducible i*CARDEL*^OE^ cells. The time is indicated in days. (**c**,**d**) i*CARDEL*^OE^ hiPSC- (**c**) and hESC-derived (**d**) cardiomyocytes differentiated under control (−DOX) or induced (+DOX) conditions. Representative images of the intensity of beating areas according to the heatmap scale in the bottom (**left**). Quantification of beating areas (center). Quantification of cTnT-positive cells by flow cytometry (**right**). (**e**,**f**) Representative plots of beating frequency (**left**) and quantification of beating rate in i*CARDEL*^OE^ hiPSC- (**e**) and hESC-derived (**f**) cardiomyocytes differentiated under control (−DOX) or induced (+DOX) conditions. (**g**,**h**) Action-potential profiles of i*CARDEL*^OE^ (**g**) hiPSC- and (**h**) hESC-derived ventricular cardiomyocytes under control (−DOX) or induced (+DOX) conditions. Mean with SD; Student’s unpaired *t*-test analysis: * *p* < 0.05, ** *p* < 0.01, *** *p* < 0.001, **** *p* < 0.0001.

**Figure 4 cells-13-01050-f004:**
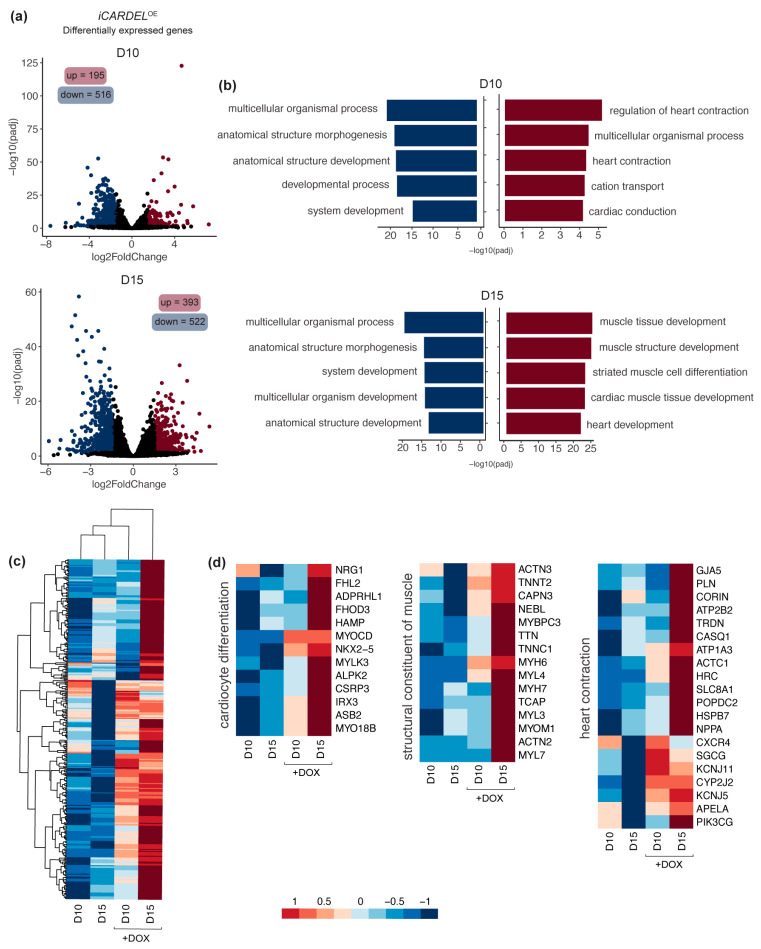
*CARDEL* contributes to molecular changes during cardiomyocyte differentiation. (**a**) RNA-seq analysis of i*CARDEL*^OE^ hiPSC-derived cardiomyocytes differentiated under control (−DOX) or induced (+DOX) conditions. Volcano plots show differentially expressed genes for −DOX versus +DOX: upregulated in red, downregulated in blue (padj < 0.05, −1.5 ≥ log2FC ≥ 1.5). (**b**) Genome ontology (GO) analysis of upregulated (red) and downregulated (blue) sets of genes. The five most representative GO terms are shown. (**c**) Heatmap and cluster of upregulated genes in D15. (**d**) Heatmap of upregulated genes in D15 related to specific GO terms, as indicated. Values are plotted as log2 of normalized counts by DESeq2.

## Data Availability

RNA-Seq data generated in this study have been deposited in the NCBI Sequence Read Archive under accession number PRJNA926378 and are publicly available as of the date of publication. RNA-seq data of in vitro cardiac differentiation was previously generated by our group [17] and is available at NCBI Sequence Read Archive, accession number SRP150416. RNA-seq data heart tissues were acquired from European Nucleotide Archive (ENA) (http://www.ebi.ac.uk/ena, accessed on 25 May 2022). Accession numbers for adult heart tissue data: ERR315356, ERR315430, ERR315367 and ERR315331. Project number for fetal heart tissue data: PRJEB27811 [18].

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
