# Peer review of "Cardiac Development Long Non-Coding RNA (CARDEL) Is Activated during Human Heart Development and Contributes to Cardiac Specification and Homeostasis"

_cells, 2024, doi:10.3390/cells13121050_

Round 1

Reviewer 1 Report

Comments and Suggestions for Authors

The authors identify CARDEL, a long-noncoding RNA, which is enriched in early human cardiomyocyte differentiation and during human fetal heart development. Subsequently, the authors generate CRISPR-targeted hiPSCs and hESCs for loss of CARDEL and demonstrate a reduction in the propensity for cardiac differentiation. Further, overexpression of CARDEL shows evidence of increasing cardiac differentiation. The authors evaluate physiologic and genomic data to support these findings. 

Minor comments:

Figure 3. By just looking at the figure panels, it is not clear to which data comes from hiPSCs or hESCs unless I read the figure legend. As done in Figure 2, the labels for hiPSC or hESC are useful.

Figure 4. The font in the figures is not sharp, nor is the heatmap in panel C.

Comments on the Quality of English Language

Minor comments:

I noticed only a very minor need for editing of English in the manuscript, notably in the Discussion.

Author Response

Reviewer 1

Minor comments:

  • Figure 3. By just looking at the figure panels, it is not clear to which data comes from hiPSCs or hESCs unless I read the figure legend. As done in Figure 2, the labels for hiPSC or hESC are useful.

We agree with the reviewer and added the suggested labels on Figure 3.

  • Figure 4. The font in the figures is not sharp, nor is the heatmap in panel C.

The figure was updated for better quality.

Reviewer 2 Report

Comments and Suggestions for Authors

In this interesting work, authors searched for uncharacterized cardiac developmental genes by combining a temporal evaluation of the human cardiac specification in vitro with the analysis of fetal and adult heart tissue gene expression. The study is well conducted,  however, it is not fully clear the aim of the work. Please specify.

How did the authors obteined H1 hESC line and IPRN13.13 iPSC line cells?

Author Response

Reviewer 2

  • The study is well conducted,  however, it is not fully clear the aim of the work. Please specify.

In accordance with the reviewer's suggestion, we added the aim of the work in the main text, lines 60-65: “Here, we aimed to develop a strategy to find and test a novel cardiac developmental gene. We combined a temporal evaluation of the human cardiac specification in vitro with an analysis of fetal and adult heart tissue gene expression to search for uncharacterized genes. This strategy led us to find CARDEL (CARdiac DEvelopment Long non-coding RNA), and we functionally characterized it using genetic approaches.”

  • How did the authors obteined H1 hESC line and IPRN13.13 iPSC line cells?

We thank the reviewer for suggesting to add this information. The main text was updated on lines 72-75: “Human embryonic stem cell, H1 line, was obtained from WiCell Research Institute (Madison, WI, USA) under a Materials Transfer Agreement (No.18-W0416) with Carlos Chagas Institute. Human induced pluripotent cell line, IPRN13.13, was generated and described by Darabi et. al, 2012 (https://doi.org/10.1016/j.stem.2012.02.015).”

Reviewer 3 Report

Comments and Suggestions for Authors

In this manuscript the authors screen for and discuss the role of lncRNA CARDEL (cardiac development long non-coding RNA) in the context of cardiomyocyte (CM) differentiation using one line of each hiPSC and hESC. The methods applied are not well described and several details are missing/unclear.

While reading through the manuscript I had the following concerns:

·         Line 322-324: the authors mention iCARDELOE hiPSC and hESC-CM compared to non-induced control. However, there are only 2 videos provided and it is unclear which video corresponds to which group

·         LncRNA CARDEL is known as LINC00890 or SERTM2. Is this lncRNA only expressed in cardiomyocytes?  Based on other reports (PMID: 28587571) in the literature this does not seem to be the case. So it is not clear to me why the authors believe that this lncRNA has a role for cardiac development. Did the authors check the expression for this lncRNA in other organs except the heart (fetal vs adult)?  

·         Scale bars are missing in Fig 2C.

·         An introduction about SERTM2 or LINC00890 will benefit the manuscript. Is it nulcear or cytoplasmic localized? What could be the potential molecular mechanisms by which it is regulating CM differentiation?

·         The authors should show pluripotency marker expression or staining for the CARDELKO hiPSC lines generated in this study. Colony morphology and karyogram are not sufficient to claim maintenance of pluripotency. Also a trilineage differentiation experiment would help to confirm if this lncRNA has a specific effect for mesodermal/ CM differentiation only or is also important in general for development.

·         Fig 2D&E- if the authors quantified low cTnT positive cells after the differentiation then it is obviously expected that the beating rate would be lower. If the authors want to show/prove the role of CARDEL for CM development then they should look at the beating rate in purified CM population. At which day of the differentiation was the beating rate calculated?

·         Line 87-89: the authors write that the doxycycline induction was performed on day 5 and day 7. It is not clear how or why they chose these specific timepoints? Did the authors check first how much OE is achieved (during the differentiation timeline)? In the current manuscript the authors show separately in Fig 1f / Suppl Fig 2b the expression of CARDEL during the differentiation in WT vs KO cells and Suppl Fig 3a the OE after Dox. But it is not clear at which time point or how many days after DOX stimulation. Please clarify

·         Fig 3c and d – the heatmap scale- colour code is missing. Please update.

Round 2

Reviewer 3 Report

Comments and Suggestions for Authors

Thank you for answering all concerns.

there is one minor issue- in Fig S3 the authors show data for ventricular, atrial and nodal CMs from the CARDEL OE conditions in hiPSC and hESC-  but there is no description in methods section as to how these CM subtypes were generated / sorted / categorized. Based on differentiation protocol mentioned in the methods section one would expect to produce majorly ventricular CMs. so how did the authors get sufficient quantities of atrial and nodal CMs or was this based on electrophysiological data generated after patch clamp experiments. please clarify and update text accordingly.

Author Response

Reviewer #3

  • there is one minor issue- in Fig S3 the authors show data for ventricular, atrial and nodal CMs from the CARDEL OE conditions in hiPSC and hESC-  but there is no description in methods section as to how these CM subtypes were generated / sorted / categorized. Based on differentiation protocol mentioned in the methods section one would expect to produce majorly ventricular CMs. so how did the authors get sufficient quantities of atrial and nodal CMs or was this based on electrophysiological data generated after patch clamp experiments. please clarify and update text accordingly.

The categorization was based on electrophysiological data from patch clamp. We did not attempt to generate cardiomyocyte subtypes. The results were a measurement we got with the differentiation protocol described. The main text was updated for clarification. Lines 351-353: “Whole cell patch clamp assay was performed in iCARDELOE hPSC-derived cardiomyocytes and allowed the identification of cardiomyocyte subtypes by their electrophysiological properties.”